# Mastectomy, HER2 Receptor Positivity, NPI, Late Stage and Luminal B-Type Tumor as Poor Prognostic Factors in Geriatric Patients with Breast Cancer

**DOI:** 10.3390/diagnostics15010013

**Published:** 2024-12-25

**Authors:** Demet Nak, Mehmet Kivrak

**Affiliations:** 1Department of Nuclear Medicine, Education and Training Hospital, Recep Tayyip Erdoğan University, 53020 Rize, Turkey; demetnak@yahoo.com; 2Department of Biostatistics and Medical Informatics, Division of Basic Medical Sciences, Faculty of Medicine, Recep Tayyip Erdoğan University, 53020 Rize, Turkey

**Keywords:** elder, breast cancer, prognosis, survival

## Abstract

**Background/Objectives:** This study aims to explore the risk factors associated with poor survival outcomes in geriatric female patients with breast cancer. **Methods:** This study utilized data from the METABRIC database to evaluate the risk factors associated with poor survival outcomes among geriatric breast cancer patients. A total of 2909 female patients, 766 of whom were geriatric, were included in the study. The effects of the type of surgery; breast cancer types; cellularity; Estrogen receptor (ER), progesterone receptor (PR), and human epidermal growth factor receptor 2 (HER2) status; molecular class; axillary lymph nodes; Nottingham prognostic index (NPI); status of receiving systemic chemotherapy (SCT), hormone therapy (HT), and radiotherapy (RT); tumor size and tumor on overall survival (OS); and progression-free status (PFS) of geriatric patients were investigated. Additionally, the disease-specific survival of geriatric patients was compared with other patients. **Results:** HER2 receptor positivity, advanced-stage tumors (T3–T4), a high NPI, and Luminal B subtypes were significant predictors of worse outcomes. Conversely, Luminal A tumors, associated with favorable hormonal responsiveness, demonstrated the best progression-free survival (PFS). HER2-positive patients exhibited a poorer PFS compared to their HER2-negative counterparts, underscoring the need for careful management of aggressive subtypes in older adults. Additionally, patients undergoing mastectomy were less likely to receive adjuvant therapies, contributing to inferior outcomes compared to breast-conserving surgery (BCS). **Conclusions:** Mastectomy, HER2 positivity, high NPI, advanced stages, and Luminal B tumors are significant prognostic factors in geriatric breast cancer patients.

## 1. Introduction

Breast cancer is the most frequently diagnosed malignancy in women worldwide and ranks as the second leading cause of cancer-related deaths in this population. According to the World Health Organization (WHO), in 2022, there were 2.3 million women diagnosed with breast cancer globally, resulting in 670,000 deaths. The occurrence of breast cancer varies widely, depending on factors such as geographic location, age, genetic predisposition, and environmental influences. Key risk factors include advanced age, a family history of the disease, hormonal factors such as early onset of menstruation or late menopause, obesity, alcohol consumption, and the use of hormonal therapies. Understanding how these factors interact is vital for evaluating an individual’s risk and shaping screening strategies [1,2,3].

The most commonly diagnosed pathological subtypes of invasive breast cancer are invasive ductal carcinoma and invasive lobular carcinoma [4,5,6]. Diagnosis involves a multidisciplinary approach, combining physical examinations, imaging techniques such as mammography and ultrasound, and confirmation through tissue biopsy. Advanced staging relies on imaging modalities like computed tomography, magnetic resonance imaging, and metabolic imaging techniques, including F18-fluorodeoxyglucose positron emission tomography (F18-FDG PET/CT) [7,8,9]. Treatment options are diverse, ranging from surgery, chemotherapy, and radiotherapy to hormonal and immunotherapies. Treatment decisions are guided by tumor characteristics, including histology, stage, receptor status, and genetic markers, alongside additional factors such as tumor grade and proliferation indices [5,6,10,11].

With increasing life expectancy and advances in early detection, breast cancer is becoming more prevalent among older adults. The risk of developing the disease rises significantly with age, from 1.5 cases per 100,000 women aged 20–24 to over 400 cases per 100,000 among those aged 75–79. Moreover, nearly half of all breast cancer-related deaths in Western countries occur in women aged 70 and above. Breast cancer comprises various biological subtypes, each characterized by different prognoses and treatment responses. However, older patients face unique challenges, such as comorbidities, side effects from treatments, and a lack of evidence-based guidelines specific to geriatric care. Addressing these research gaps is essential for improving outcomes for this vulnerable patient group [12,13,14,15,16,17].

This study aims to evaluate the factors influencing poor survival outcomes in elderly patients with breast cancer, providing insights that can inform clinical practice and direct future research.

## 2. Material and Methods

### 2.1. Dataset Description

This study utilized data from the Molecular Taxonomy of Breast Cancer International Consortium (METABRIC) database. The dataset, collected by researchers from Cambridge Research Institute and British Columbia Cancer Centre (GB), includes variables such as age at diagnosis, surgery type (mastectomy or breast-conserving surgery, BCS), breast cancer subtypes (invasive ductal carcinoma, invasive lobular carcinoma, mixed ductal and lobular carcinoma, and invasive breast carcinoma), receptor statuses (ER, PR, and HER2), molecular subtypes (Luminal A, Luminal B, HER2-enriched, and basal), Nottingham prognostic index (NPI), menopausal status, lymph node involvement, and therapeutic modalities (systemic chemotherapy [SCT], hormone therapy [HT], and radiotherapy [RT]). Additionally, tumor size, stage, overall survival (OS), progression-free survival (PFS), and cause of death were included. In order to determine the differences between geriatric patients and other patient groups according to their age and menopausal status, the patients were divided into four groups: geriatric (≥65 years, *n* = 766), non-geriatric menopausal (*n* = 681), premenopausal (*n* = 392), and all non-geriatric patients (*n* = 1070). The OS and PFS were calculated over 5 years, with breast cancer-specific mortality separated from other causes of death.

### 2.2. Statistical Analyses

Continuous variables with normal distribution were summarized as the mean ± standard deviation, while those without normal distribution were summarized as the median (IQR). Categorical variables were presented as counts (percentage). The Kolmogorov–Smirnov test was used to assess the normality assumption. Since the continuous variables followed a normal distribution, the one-way ANOVA test was applied. For categorical variables, the chi-square test or Fisher’s exact test was used for independent groups. Pairwise comparisons were conducted using Bonferroni-adjusted post hoc tests. Survival analyses for the overall survival and progression-free survival values were performed using the Kaplan–Meier method. A significance level of 0.05 was set for all statistical tests. Statistical analyses were conducted using SPSS version 29.0.2.0 and Jamovi version 2.4.6 software.

## 3. Results

### 3.1. Demographics and Tumor Characteristics

The mean ages were 73.4 ± 5.86, 57.9 ± 4.28, 42.6 ± 5.57, and 52.3 ± 8.8 years for geriatric, non-geriatric menopausal, premenopausal, and all non-geriatric groups, respectively. Invasive ductal carcinoma predominated across the groups (80.1%), and Luminal A was the most frequent molecular subtype. ER/PR positivity and HER2 negativity were common. Tumor stage analysis revealed stage 2 as the most prevalent and stage 4 as the least common (*p* < 0.001). Geriatric patients had a higher proportion of mastectomy cases (67.4%, *p* < 0.001), while BCS was more common in younger groups (Table 1 and Table 2).

Geriatric patients had significantly lower OS rates compared to all other groups (log-rank *p* < 0.001, Figure 1, Table 3). Their disease-specific mortality risk was 1.96–2.33 times higher in geriatric patients than in other groups (*p* < 0.001). The PFS times were not statistically different among the groups (log-rank *p* = 0.37), with geriatric patients showing a mean PFS of 170 ± 6.06 months. Non-cancer-related deaths were markedly more common in geriatric patients (55%) compared to non-geriatric patients (4.5%, *p* < 0.01).

### 3.2. Geriatric Cohort Outcomes

Among geriatric patients, HER2-negative individuals exhibited significantly longer PFSs (184 ± 5.35 months) than HER2-positive patients (134 ± 20.33 months, *p* = 0.0095, Figure 2). However, these subgroups had no significant difference in OS (log-rank *p* = 0.076). The mean NPI was significantly higher in geriatric patients with disease progression (4.42 ± 0.95) compared to those without progression (3.87 ± 0.83, *p* < 0.001, Figure 2).

Mastectomy was more common among geriatric patients than BRC (67.4% vs. 32.6%, *p* < 0.001), and the survival outcomes in geriatric patients varied by surgical type: BCS was associated with a superior 5-year OS (233 ± 11.59 months vs. 204 ± 7.91 months for mastectomy) and PFS (202 ± 8.56 months vs. 170 ± 6.06 months for mastectomy, *p* < 0.01) (Figure 2). Geriatric patients undergoing mastectomy were significantly more likely to forgo SCT and radiotherapy RT (*p* < 0.001). Among those who underwent mastectomy, 65.4% did not receive SCT and RT, compared to only 13.1% in the BCS group (Table 4). The rates of receiving and not receiving HT were similar in both surgery types.

Early-stage (T1/T2) geriatric patients had a substantially better OS (131.9 ± 3.35 months) and PFS (185 ± 5.34 months) than late-stage (T3/T4) patients, whose OS and PFS were 87.4 ± 9.66 months and 124 ± 19.78 months, respectively (*p* < 0.001, Figure 2).

In the multivariate analysis of OS, 93.3% (*n* = 715) of patients were in the early stage of the disease, while 6.7% (*n* = 51) were in the late stage. Regarding surgery types, 32.6% (*n* = 250) underwent BCS, and 67.4% (*n* = 516) underwent mastectomy. The mean NPI was 4 ± 1.1. The model demonstrated a concordance index of 0.62 (SD: 0.012) and an R-squared value of 0.82. The likelihood ratio chi-square test yielded a value of 65.58 (df = 3, *p* < 0.01).

In the multivariate analysis of PFS, 92.3% (*n* = 707) of patients had a HER2-negative status, and 7.7% (*n* = 59) had a HER2-positive status. Regarding surgery types, 32.6% (*n* = 250) underwent BCS, and 67.4% (*n* = 516) underwent mastectomy. Similar to the OS, 93.3% (*n* = 715) of patients were in the early stage, and 6.7% (*n* = 51) were in the late stage. The model demonstrated a concordance index of 0.579 (SD: 0.016) and an R-squared value of 0.39. The likelihood ratio chi-square test yielded a value of 30.14 (df = 3, *p* < 0.01).

Luminal A patients had the longest PFS (200 ± 7.71 months), while Luminal B patients had the shortest PFS (159 ± 8.12 months, log-rank *p* = 0.0024). Triple-negative patients exhibited intermediate PFS (187 ± 13.3 months, Figure 3). The OS times did not differ significantly among molecular subtypes (*p* = 0.72).

Mortality due to non-breast cancer causes differed significantly between geriatric (*n* = 329) and non-geriatric (*n* = 131) patients in relation to the treatment received. Geriatric patients were less likely to have undergone systemic chemotherapy or radiotherapy compared to non-geriatric patients (*p* = 0.016), potentially reflecting age-related differences in treatment eligibility or comorbidities (Table 5).

There was no significant difference in the OS and PFS among patients with different histopathological subtypes. The OS and PFS rates were similar between the ‘cellularity high’ and ‘cellularity low’ groups. No significant differences were observed between claudin-low and other triple-negative breast cancer subtypes in terms of the OS and PFS. The ER-positive/negative and PR-positive/negative groups exhibited no significant differences in OS or PFS. No significant correlation was found between examined axillary lymph nodes and the OS/PFS of geriatric patients.

## 4. Discussion

Breast cancer in geriatric patients presents a significant clinical challenge due to the interplay not only by complex biological and age-related medical factors but also by the unique social and physiological characteristics of older patients that influence disease outcomes and treatment decisions. This study identifies mastectomy, HER2 receptor positivity, high NPI, advanced-stage tumors, and Luminal B molecular subtypes as significant poor prognostic factors in geriatric breast cancer patients.

The association between mastectomy and poor survival outcomes may reflect the systemic challenges in managing elderly patients. Geriatric patients who undergo mastectomy are less likely to receive adjuvant therapies such as SCT and RT. This may be due to concerns about the tolerability of SCT and RT comorbidities or functional decline in elderly populations, which may contribute to their inferior survival outcomes compared to patients treated with breast-conserving surgery (BCS) [15,16]. This finding is consistent with evidence that treatment decisions in older adults are often shaped by factors such as frailty, comorbidities, and functional decline, emphasizing the complexities of treating older patients with cancer, where balancing the benefits of aggressive treatment against the risks of adverse effects requires careful consideration [15,17]. Additionally, inequities in access to health care, socioeconomic barriers, and limited care support further compound the challenges faced by this group [18].

HER2 positivity has emerged as an important prognostic factor, consistent with its role as an indicator of aggressive tumor behavior. Although HER2-targeted therapies, particularly trastuzumab, have significantly improved outcomes in younger populations, their use in older patients is limited by concerns about cardiotoxicity and other age-related complications [19]. Older patients often have pre-existing cardiac conditions, such as hypertension or ischemic heart disease, which increase the risk of treatment-induced cardiac dysfunction, leading to the underutilization of these therapies. Studies have emphasized that these risks can be reduced and tolerability can be increased in elderly patients with strategies such as dose adjustments and cardiac monitoring [15,20,21,22]. However, inequities in access to new agents such as trastuzumab-deruxtecan remain a concern [22].

Age-related factors, including vulnerability, multiple medication use, and reduced organ function, further complicate the administration of these treatments, while systemic barriers such as limited access to care exacerbate disparities. Consequently, the prognosis for HER2-positive geriatric patients remains poor compared to younger populations [14,23,24,25,26]. Despite these challenges, emerging evidence suggests that careful patient selection, cardiac monitoring, and dose adjustments can allow for the safe and effective use of HER2-targeted therapies in older patients [27]. To address this gap, comprehensive geriatric assessments are crucial for evaluating patients’ overall health and social factors, enabling personalized treatment approaches. Additionally, newer agents with more favorable safety profiles, such as trastuzumab-deruxtecan, may expand treatment options for this population [15]. Promoting access to HER2-targeted treatments and further research into geriatric-specific treatment strategies are essential to improve the outcomes for HER2-positive geriatric breast cancer patients.

Advanced disease and high NPI scores once again emphasize the importance of early diagnosis in improving outcomes. A delay in diagnosis in elderly patients due to reasons such as decreased participation in routine screening programs or attribution of symptoms to aging is a significant problem [2]. Social isolation, prevalent among older adults, can contribute to delays in seeking medical attention, leading to more advanced disease at diagnosis or adhering to treatment regimens, compounding their already significant prognostic disadvantage. These observations are consistent with global cancer statistics, underscoring the crucial role of early detection in improving outcomes and emphasizing the importance of screening and early intervention in reducing cancer-related mortality [1,24].

The molecular subtype analysis revealed that Luminal A represents the most prevalent molecular subtype among geriatric breast cancer patients, consistent with the findings of a previous study, which also reported a predominance of this subtype in older populations [24]. Furthermore, the observation that HER2-positive tumors are the least common within this demographic aligns with prior studies. The high prevalence of ER+ and PR+ tumors, as demonstrated in this analysis, is similarly in agreement with earlier literature, emphasizing the predominance of hormone receptor-positive subtypes in older patients, which are often characterized by lower proliferation rates and improved responsiveness to endocrine therapies.

Of note, Luminal A tumors, which are more hormone-responsive, were associated with favorable outcomes, whereas Luminal B subtypes exhibited significantly worse survival. This distinction highlights the need for molecular profiling to guide treatment decisions, particularly given the challenges of managing comorbidities and treatment-related toxicity in older patients [25,26]. Despite its value, molecular profiling has been noted to be underutilized in this population, highlighting disparities in care delivery. Access to advanced diagnostic tools needs to be increased in this population [27].

Interestingly, in contrast to the findings in younger populations, traditional markers such as axillary lymph node involvement and ER/PR receptor status were not found to be significant predictors of survival among geriatric patients. Instead, tumor size (T3–T4) and HER2 positivity emerged as critical prognostic indicators, corroborating the findings of a study that similarly highlighted their association with poorer prognoses in elderly cohorts [25]. Insignificance of the effect of axillary lymph node involvement, histopathological subtypes, cellularity, and ER/PR receptor status on survival suggests that traditional pathological markers may play a secondary role in geriatric patients, where social, functional, and systemic factors often outweigh purely biological determinants [28].

Older patients with breast cancer often present with significant physiological vulnerabilities, including comorbidities, reduced organ function, and diminished physiological reserve, which may limit their ability to tolerate aggressive treatments. For instance, the association between mastectomy and poorer survival outcomes in geriatric patients may partly reflect the reluctance to offer systemic chemotherapy or radiotherapy to older individuals due to concerns about treatment-related toxicity. Additionally, functional impairments such as fragility, cognitive decline, and decreased mobility may hinder their adherence to complex treatment regimens, further exacerbating disparities in care [15].

These findings collectively reinforce the necessity of tailored therapeutic strategies that account for the unique molecular profiles and clinical characteristics of geriatric breast cancer patients, thereby optimizing outcomes in this vulnerable population. Importantly, our analysis suggests that non-breast cancer mortality in geriatric patients may reflect the influence of underlying comorbidities, complications, or functional decline, which may limit treatment initiation or completion. Geriatric patients who died from causes unrelated to breast cancer were significantly less likely to have received SCT or RT compared to their non-geriatric counterparts. This disparity underscores the potential impact of age-related health factors and treatment tolerability on outcomes. While direct data on comorbidities were not available in our dataset, these findings indirectly highlight the critical role of age-related vulnerabilities in shaping treatment decisions and overall prognosis. Future research incorporating comprehensive geriatric assessments, including comorbidities and functional status, is essential to elucidate these relationships further. Managing breast cancer in older adults requires balancing oncological outcomes with age-related vulnerabilities. Fit, older individuals can often tolerate standard treatments and achieve outcomes comparable to younger patients, while those with frailty or susceptibility may require adjusted regimens and supportive interventions [15]. Additionally, competing mortality risks, even without multimorbidities, necessitate treatment strategies that address both breast cancer recurrence risk and non-cancer mortality, which is heavily influenced by frailty. This study highlights the 55% non-cancer mortality rate among geriatric patients, alongside higher breast cancer-specific mortality compared to younger cohorts, emphasizing the need for personalized approaches [14]. Collaboration between oncologists and geriatricians is essential to address the complex needs of this population to guide treatment decisions, ensuring fragile individuals receive tailored strategies that focus on supportive care and quality of life [27]. Future research should focus on incorporating these assessments with molecular profiling to refine prognostic models further and develop socially and biologically informed therapeutic strategies.

This study has several limitations, including incomplete TNM staging data, which restricts a comprehensive evaluation of tumor burden and disease progression. The absence of detailed nodal and metastasis (N and M) information limits the understanding of their impact on survival outcomes. Key variables such as BRCA mutation status, molecular profiling, and treatment adherence data were insufficiently reported, hindering a full assessment of genetic, molecular, and therapeutic influences on prognosis. Additionally, the lack of patient-related factors such as comorbidities, functional status, and socioeconomic determinants reduces the ability to address the unique needs of geriatric populations. Finally, the underrepresentation of certain breast cancer subtypes, such as triple-negative breast cancer, limits the generalizability of these findings.

## 5. Conclusions

In conclusion, mastectomy, HER2 positivity, high NPI, advanced-stage, and Luminal B tumors are significant prognostic factors in geriatric breast cancer patients. However, the outcomes in this population are deeply influenced by age-related medical vulnerabilities and social limitations. Navigating the complexities of cancer care and effective management for older adults requires a holistic approach that accounts for the interplay of complex biological, physiological, and social determinants of health, ultimately aiming to improve outcomes, personalized care, and quality of life for this vulnerable population.

## Figures and Tables

**Figure 1 diagnostics-15-00013-f001:**
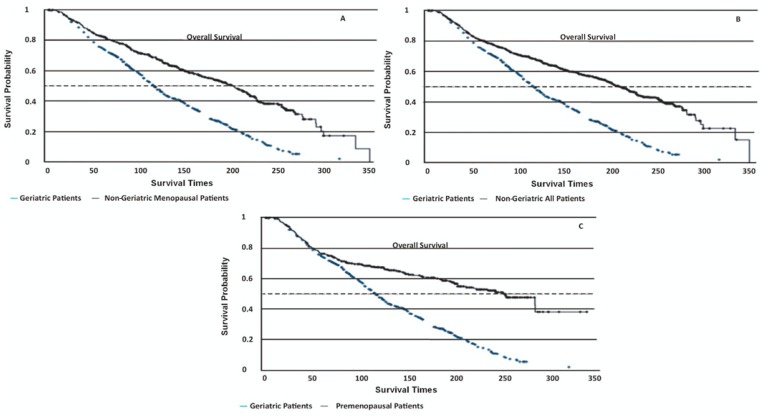
Kaplan–Meier survival curves depicting disease-specific overall survival (OS) among different patient groups. (**A**) Geriatric patients: 130 ± 3.22 months (95% CI 108–154); non-geriatric menopausal patients: 191 ± 5.41 months (95% CI 182–218) (*p* < 0.001). (**B**) Geriatric patients: 130 ± 3.22 months (95% CI 108–154); non-geriatric all patients: 198 ± 4.74 months (95% CI 196–226) (*p* < 0.001). (**C**) Geriatric patients: 130 ± 3.22 months (95% CI 108–154); premenopausal patients: 214 ± 8.36 months (95% CI 202–249) (*p* < 0.001).

**Figure 2 diagnostics-15-00013-f002:**
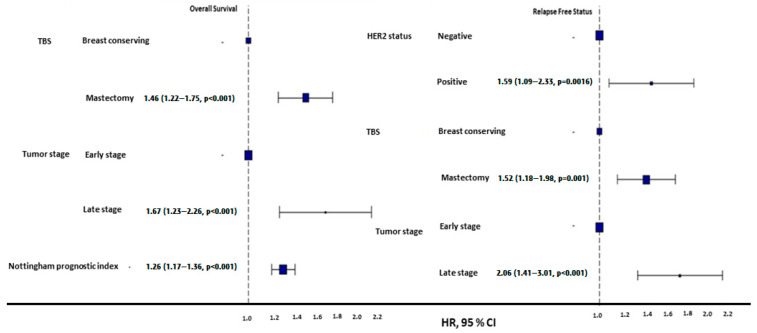
HR plots of overall and progression-free survival in geriatric patients. Mastectomy was associated with poorer OS and PFS compared to breast-conserving surgery. Late-stage tumors demonstrated significantly worse OS and PFS outcomes compared to early-stage tumors. HER2-positive tumors were linked to higher risks of progression compared to HER2-negative tumors. Additionally, a higher Nottingham prognostic index (NPI) correlated with a poorer OS.

**Figure 3 diagnostics-15-00013-f003:**
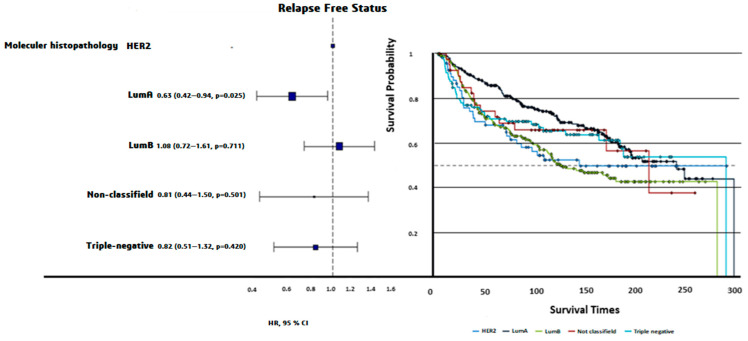
HR plot and progression-free survival curves of PAM50 molecular subtypes. Luminal A demonstrated the longest PFS (200 ± 7.71 months), followed by triple-negative (187 ± 13.3 months), nonclassified (181 ± 23.25 months), HER2-enriched (173 ± 16.41 months), and Luminal B (159 ± 8.12 months). Differences in PFS across molecular subtypes were statistically significant (Log-rank *p* = 0.0024).

**Table 1 diagnostics-15-00013-t001:** Descriptive histopathological characteristics of patient groups.

Group	Geriatric Patients (*n* = 766)	Non-Geriatric Menopausal Patients (*n* = 681)	Premenopausal Patients (*n* = 392)	Non-Geriatric All Patients (*n* = 1070)		
	Descriptive Statistics				Test Statistics	*p*-Value
**Cancer Type Detailed**						
Invasive Ductal Carcinoma	587 (76.6%)	535 (78.6%)	336 (85.7%)	869 (81.2%)	16.49	0.057 *
Invasive Lobular Carcinoma	64 (8.4%)	57 (8.4%)	20 (5.1%)	77 (7.2%)		
Mixed Ductal and Lobular Carcinoma	95 (12.4%)	71 (10.4%)	31 (7.9%)	102 (9.5%)		
Invasive Breast Carcinoma	20 (2.6%)	18 (2.6%)	5 (1.3%)	22 (2.1%)		
**ER Status**						
Negative	100 (13.1%)	164 (24.1%)	165 (42.1%)	328 (30.7%)	132.75	**<0.001 ***
Positive	666 (86.9%)	517 (75.9%)	227 (57.9%)	742 (69.3%)		
**PR Status**						
Negative	321(41.9%)	341 (50.1%)	201 (51.3%)	541 (50.6%)	17.0	**<0.001 ***
Positive	445 (58.1%)	340 (49.9%)	191 (48.7%)	529 (49.4%)		
**HER2 Status**						
Negative	707 (92.3%)	586 (86.1%)	321 (81.9%)	905 (84.6%)	32.86	**<0.001 ***
Positive	59 (7.7%)	95 (13.9%)	71 (18.1%)	165 (15.4%)		
**Cellularity**						
High	398 (51.9%)	342 (50.2%)	197 (50.3%)	539 (50.4%)	4.79	0.57 *
Moderate	294 (38.4%)	267 (39.2%)	142 (36.2%)	407 (38.1%)		
Low	74 (9.7%)	72 (10.6%)	53 (13.5%)	124 (11.5%)		
**Molecular Classification**						
HER2	68_a_ (8.9%)	88_a_ (12.9%)	48_a_ (12.2%)	135_b_ (12.6%)	141.0	**<0.001**
Luminal A	308_b_ (40.2%)	242_b_ (35.5%)	122_b_ (31.1%)	363_a_ (33.9%)		
Luminal B	243_c_ (31.7%)	157_c_ (23.1%)	46_b_ (11.7%)	203_c_ (19.0%)		
Triple-Negative	106_d_ (13.8%)	140_d_ (20.6%)	133_c_ (34.0%)	272_d_ (25.4%)		
Nonclassified	41_d_ (5.4%)	54_e_ (7.9%)	43_c_ (11.0%)	97_e_ (9.1%)		

Different letters in the columns indicate significant differences between categories. *: Chi-square Fisher exact test in independent groups.

**Table 2 diagnostics-15-00013-t002:** Descriptive disease and treatment characteristics of patient groups.

Group	Geriatric Patients (*n* = 766)	Non-Geriatric Menopausal Patients (*n* = 681)	Premenopausal Patients (*n* = 392)	Non-Geriatric All Patients (*n* = 1070)		
**Tumor Stage**						
Stage 1	269_a_ (35.4%)	332_a_ (48.8%)	181_b_ (46.6%)	512_c_ (48.1%)	37.4	**<0.001 ***
Stage 2	444_b_ (58.5%)	313_a_ (46.0%)	186_b_ (47.9%)	497_c_ (46.7%)		
Stage 3	42_c_ (5.5%)	31_b_ (4.6%)	20_c_ (5.2%)	51_a_ (4.8%)		
Stage 4	4_d_ (0.5%)	5_c_ (0.6%)	5_d_ (0.3%)	10_b_ (0.5%)		
**Type of Breast Surgery**						
Breast-Conserving	250 (32.6%)	314 (46.1%)	165 (44.6%)	486 (45.4%)	38.51	**<0.001 ***
Mastectomy	516 (67.4%)	367 (53.9%)	227 (55.4%)	584 (54.6%)		
**Nottingham prognostic index**	4.0004 ± 1.1314	4.0414 ± 1.1395	4.1181 ± 1.1309	4.2506 ± 1.1027	4.883	0.002 **
**Systemic Chemotherapy**						
Yes	730 (95.3%)	515 (75.6%)	203 (51.8%)	717 (67.0%)	312.0	**<0.001 ***
No	36 (4.7%)	166 (24.4%)	189 (48.2%)	353 (33.0%)		
**Hormone Therapy**						
Yes	200 (26.1%)	239 (35.1%)	253 (64.5%)	492 (46.0%)	181.0	**<0.001 ***
No	566 (73.9%)	442 (64.9%)	139 (35.5%)	578 (54.0%)		
**Radio Therapy**						
Yes	359 (46.9%)	239 (35.1%)	129(32.9%)	367 (34.3%)	38.5	**<0.001 ***
No	407 (53.1%)	442 (64.9%)	263 (67.1%)	703 (65.7%)		

Different letters in the columns indicate significant differences between categories. *: Chi-square Fisher exact test in independent groups. **: One-way ANOVA.

**Table 3 diagnostics-15-00013-t003:** Pairwise comparisons of overall survival differences between patient groups.

Groups	Levels	*p*-Value
Non-Geriatric All Patients	Geriatric Patients	**<0.001**
Non-Geriatric Menopausal Patients	Geriatric Patients	**<0.001**
Non-Geriatric Menopausal Patients	Non-Geriatric All Patients	0.502
Premenopausal Patients	Geriatric Patients	**<0.001**
Premenopausal Patients	Non-Geriatric All Patients	0.502
Premenopausal Patients	Non-Geriatric Menopausal Patients	0.283

Note. *p*-value adjustment method: Bonferroni.

**Table 4 diagnostics-15-00013-t004:** Systemic chemotherapy and radiotherapy treatment data of patients according to surgery types.

		Geriatric Patients (*n* = 766)		Non-Geriatric All Patients (*n* = 1070)			
Group		Breast-Conserving (*n* = 250)	Mastectomy (*n* = 516)	Breast-Conserving (*n* = 486)	Mastectomy (*n* = 584)		
Chemotherapy	Radiotherapy	Descriptive Statistics		Descriptive Statistics		Test Statistics	*p*-Value
Not received	Not received	32 (13.1%)	318 (65.4%)	26(7.3%)	278 (76.6%)	351.8	**<0.001**
	Received	212 (86.9%)	168 (34.6%)	328 (%92.7)	85 (23.4%)		
Received	Not received	0 (0.0%)	9 (30.0%)	7 (5.3%)	56 (25.3%)	22.6	**<0.001**
	Received	6 (%100)	21 (70.0%)	125 (94.7%)	165 (74.7%)		

**Table 5 diagnostics-15-00013-t005:** Patients died from causes unrelated to breast cancer.

	Geriatric Patients (*n* = 329)	Non-Geriatric All Patients (*n* = 131)		
Therapy	Descriptive Statistics		Test Statistics	*p*-Value
Received neither SCT nor RT	179 (54.4%)	55 (41.9%)	5.79	**0.016**
Received either or both SCT and RT	150 (45.6%)	76 (58.1%)		

## Data Availability

The data presented in this study are openly available in https://www.kaggle.com/datasets/gunesevitan/breast-cancer-metabric (accessed on 3 Novenber 2024).

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
