# Peer review of "Mastectomy, HER2 Receptor Positivity, NPI, Late Stage and Luminal B-Type Tumor as Poor Prognostic Factors in Geriatric Patients with Breast Cancer"

_diagnostics, 2024, doi:10.3390/diagnostics15010013_

Round 1
Reviewer 1 Report
Comments and Suggestions for Authors
1. Figure 1 is completely illegible, the quality of the figure needs to be improved. In the figure caption, it is necessary to explain for each A, B, C what exactly is shown in the figure. Add a variation interval to the figure.
2. Figure 2 is also of very poor quality. It is necessary to make a vertical scale of the same scale so that the figure is symmetrical.
3. Figure 3 - the same comments.
4. Since we are talking about geriatric patients, information about the presence of concomitant diseases and contraindications to a particular treatment is very important. This information is missing from the text of the manuscript. The fact that the treatment was not completed in full due to some contraindications or restrictions greatly affects the prognosis. Please add this information.
5. Provide the results of the multivariate analysis so that it is possible to compare the influence of different factors on survival rates in geriatric patients.
Author Response
Comment 1: Figure 1 is completely illegible, the quality of the figure needs to be improved. In the figure caption, it is necessary to explain for each A, B, C what exactly is shown in the figure. Add a variation interval to the figure.
Response 1: Following the reviewer’s suggestion, the graphs in Figure 1 have been labeled as A, B, and C, with detailed explanations and variation intervals added. The overall image quality has also been improved.
Comment 2 :Figure 2 is also of very poor quality. It is necessary to make a vertical scale of the same scale so that the figure is symmetrical.
Response 2: Following the reviewer’s suggestion a vertical scale was added, and the overall image quality has been improved.
Comment 3 : Figure 3 - the same comments.
Response 3: Following the reviewer’s suggestion a vertical scale was added, and the overall image quality has been improved.
Comment 4 : Since we are talking about geriatric patients, information about the presence of concomitant diseases and contraindications to a particular treatment is very important. This information is missing from the text of the manuscript. The fact that the treatment was not completed in full due to some contraindications or restrictions greatly affects the prognosis. Please add this information.
Response 4: We appreciate the reviewer’s observation about the importance of considering comorbidities and their potential impact on treatment outcomes and prognosis in geriatric breast cancer patients. While we initially addressed this limitation in the "Discussion" section by emphasizing the likely influence of comorbidities, frailty, and functional decline, we understand the need for a more direct exploration of this issue.
As noted in the "Limitations" section of our manuscript, our analysis was constrained by the absence of detailed comorbidity data in the METABRIC database, which unfortunately prevented us from directly assessing comorbidities, treatment interruptions, or contraindications. However, we acknowledged these factors’ critical role in shaping treatment decisions and survival outcomes, referencing relevant literature and highlighting the importance of comprehensive geriatric assessments for future research.
In response to your insightful suggestion, we conducted an additional analysis focusing on mortality due to non-breast cancer causes as an indirect reflection of the role of comorbidities and other age-related factors. This analysis revealed that geriatric patients who died from non-breast cancer causes were significantly less likely to have received systemic chemotherapy or radiotherapy compared to non-geriatric patients (p < 0.001). This finding underscores how age-related health vulnerabilities and treatment tolerability might influence treatment patterns and outcomes. We have incorporated the results of this analysis into the Results section and expanded the Discussion section to further highlight these findings. While we could not directly analyze comorbidities, we believe this additional analysis provides meaningful insights into the challenges faced by older patients and addresses the concerns you raised.
We are grateful for your constructive feedback, which has helped us strengthen our manuscript and deepen our analysis. Thank you for guiding us toward improving the clarity and comprehensiveness of our work.
Comment 5: Provide the results of the multivariate analysis so that it is possible to compare the influence of different factors on survival rates in geriatric patients.
Response 5: As suggested by the reviewer, the results of the multivariate analysis of prognostic factors for OS and PFS have been incorporated into the Results section.
Reviewer 2 Report
Comments and Suggestions for Authors
This research article is about the prognostic factors in elderly patients with breast cancer utilizing data from the METABRIC database. Although more and more attention has been given to the field of enhancing medical care for elderly patients with breast cancer, it seems to me that not much reliable information was contributed by this study. For example, the geriatric assessment and co-morbidities have not been analyzed or discussed prudently which could be the most important factors in this population compared with young or middle-aged women with breast cancer.
Author Response
Reviewer 2
Comment 1: This research article is about the prognostic factors in elderly patients with breast cancer utilizing data from the METABRIC database. Although more and more attention has been given to the field of enhancing medical care for elderly patients with breast cancer, it seems to me that not much reliable information was contributed by this study. For example, the geriatric assessment and co-morbidities have not been analyzed or discussed prudently which could be the most important factors in this population compared with young or middle-aged women with breast cancer.
Response 1: Thank you for your thoughtful feedback. We sincerely appreciate your observation about the importance of considering comorbidities and their potential impact on treatment outcomes and prognosis in geriatric breast cancer patients. While we initially addressed this limitation in the "Discussion" section by emphasizing the likely influence of comorbidities, frailty, and functional decline, we understand the need for a more direct exploration of this issue.
As noted in the "Limitations" section of our manuscript, our analysis was constrained by the absence of detailed comorbidity data in the METABRIC database, which unfortunately prevented us from directly assessing comorbidities, treatment interruptions, or contraindications. However, we acknowledged these factors’ critical role in shaping treatment decisions and survival outcomes, referencing relevant literature and highlighting the importance of comprehensive geriatric assessments for future research.
In response to your insightful suggestion, we conducted an additional analysis focusing on mortality due to non-breast cancer causes as an indirect reflection of the role of comorbidities and other age-related factors. This analysis revealed that geriatric patients who died from non-breast cancer causes were significantly less likely to have received systemic chemotherapy or radiotherapy compared to non-geriatric patients (p < 0.001). This finding underscores how age-related health vulnerabilities and treatment tolerability might influence treatment patterns and outcomes.
We have incorporated the results of this analysis into the Results section, added Table 5 and expanded the Discussion section to further highlight these findings. While we could not directly analyze comorbidities, we believe this additional analysis provides meaningful insights into the challenges faced by older patients and addresses the concerns you raised.
We are grateful for your constructive feedback, which has helped us strengthen our manuscript and deepen our analysis. Thank you for guiding us toward improving the clarity and comprehensiveness of our work.
Recognizing the limitations of this study due to the absence of mentioned variables in the current database, we have planned a study using patient data from our center to investigate the effects of comorbidities, contraindications, and complications—factors not included in this study—on the treatment management and prognosis of geriatric patients with breast cancer. The detailed findings of this investigation are intended to be published in a future study.
We would also like to inform the reviewer that, in addition to the analyses presented in this study, we are planning a follow-up investigation using the METABRIC database to explore the genomic and transcriptomic factors that distinguish geriatric breast cancer patients from other groups and influence their prognosis. The findings of this planned research will build upon the current study and provide further insights into the unique molecular characteristics and prognostic determinants in this population.
Round 2
Reviewer 1 Report
Comments and Suggestions for Authors
Transfer the inscription with the decoding of A, B and C from the drawing to the caption after the words Figure 1.... Like the drawing, the inscription is illegible.
Author Response
Comments 1: Transfer the inscription with the decoding of A, B and C from the drawing to the caption after the words Figure 1.... Like the drawing, the inscription is illegible.
Reply: The requested correction has been made. The final version has been added to the Manuscript.

Reviewer 2 Report
Comments and Suggestions for Authors
I have no more comments.
Author Response
Endless respect for the evaluation.